# Validation of Three MicroScan^®^ Antimicrobial Susceptibility Testing Plates Designed for Low-Resource Settings

**DOI:** 10.3390/diagnostics12092106

**Published:** 2022-08-30

**Authors:** Jean-Baptiste Ronat, Saoussen Oueslati, Alessandra Natale, Thomas Kesteman, Wael Elamin, Céline Langendorf, Liselotte Hardy, Olivier Vandenberg, Thierry Naas

**Affiliations:** 1Médecins Sans Frontières, 75019 Paris, France; 2Team ReSIST, INSERM U1184, School of Medicine University Paris-Saclay, 94270 Le Kremlin-Bicêtre, France; 3Bacteriology-Hygiene Unit, Assistance Publique-Hôpitaux de Paris, Bicêtre Hospital, 94270 Le Kremlin-Bicêtre, France; 4Oxford University Clinical Research Unit, Hanoi 10000, Vietnam; 5Institute of Health Science Education, Queen Mary University, London CB105, UK; 6Department of Microbiology, Elrazi University, GH6V East Azhari Road, Khartoum North, Sudan; 7Epicentre, 75019 Paris, France; 8Unit Tropical Bacteriology, Department of Clinical Sciences, Institute of Tropical Medicine, 2000 Antwerp, Belgium; 9Center for Environmental Health and Occupational Health, School of Public Health, Université Libre de Bruxelles (ULB), 1050 Brussels, Belgium; 10Innovation and Business Development Unit, Laboratoire Hospitalier Universitaire de Bruxelles (LHUB-ULB), Université Libre de Bruxelles (ULB), 1050 Brussels, Belgium; 11Division of Infection and Immunity, University College London, London WC1E 6BT, UK

**Keywords:** antibiotic susceptibility testing, low-resource settings, clinical bacteriology, MicroScan

## Abstract

Easy and robust antimicrobial susceptibility testing (AST) methods are essential in clinical bacteriology laboratories (CBL) in low-resource settings (LRS). We evaluated the Beckman Coulter MicroScan lyophilized broth microdilution panel designed to support Médecins Sans Frontières (MSF) CBL activity in difficult settings, in particular with the Mini-Lab. We evaluated the custom-designed MSF MicroScan Gram-pos microplate (MICPOS1) for *Staphylococcus* and *Enterococcus* species, MSF MicroScan Gram-neg microplate (MICNEG1) for Gram-negative bacilli, and MSF MicroScan Fastidious microplate (MICFAST1) for Streptococci and *Haemophilus* species using 387 isolates from routine CBLs from LRS against the reference methods. Results showed that, for all selected antibiotics on the three panels, the proportion of the category agreement was above 90% and the proportion of major and very major errors was below 3%, as per ISO standards. The use of the Prompt inoculation system was found to increase the MIC and the major error rate for some antibiotics when testing Staphylococci. The readability of the manufacturer’s user manual was considered challenging for low-skilled staff. The inoculations and readings of the panels were estimated as easy to use. In conclusion, the three MSF MicroScan MIC panels performed well against clinical isolates from LRS and provided a convenient, robust, and standardized AST method for use in CBL in LRS.

## 1. Introduction

Antimicrobial resistance (AMR) is today universally recognized as a global threat, because of the rapid emergence and dissemination of resistant bacteria and genes among humans, animals, and the environment on a global scale. It represents a heavy burden for healthcare systems all over the world [1,2,3,4]; however, the situations in low- and middle-income countries (LMICs) are particularly concerning because of the limited availability of diagnostic/surveillance and clinical/control resources [5,6].

### 1.1. Antibiotic Susceptibility Testing in Low-Resource Settings

Developing evidence-based treatment guidelines and measuring the impacts of AMR control efforts require representative and comparable data for drug-resistant bacterial infections [7]. Yet, such data have proven extremely difficult to obtain in LMICs, despite the increasing evidence that AMR is rapidly escalating in these contexts [4,8,9,10,11,12]. Available data on AMR in LMICs lack standardized laboratory and data collection practices and are often not representative of populations outside of the main cities [13]. The limited access to adequate laboratory support, in some settings with low resources, contributes to the increase in antibiotic resistance and complicates the management of infections [14]. AMR poses a unique threat in LMICs, with the potential to reverse recent progress toward infectious disease control, to damage healthcare provision generally, and threaten the safety of essential health services, such as surgery to the most vulnerable and underserved populations [8,9,12].

In this article, we define low-resource setting(s) (LRS) as an area within a country with limited access to medication, equipment, supplies, and devices, with less-developed infrastructure (electrical power, transportation, controlled environment/buildings), fewer or less-trained laboratory personnel, basic diagnostic laboratory, no expert microbiologist, and with no (or hardly introduced) clinical bacteriology [15,16,17].

The deployment of conventional microbiology laboratories in LRS is challenging. It requires complex infrastructure, logistics, equipment, and specialized human resources, often lacking in LRS [17,18,19,20,21]. Affordable and effective point-of-care (POC) diagnostics, especially those that distinguish between viral and bacterial infections, identify pathogens, and provide antibiotic susceptibility testing (AST) are yet to materialize [12,22]. As a result, Médecins Sans Frontières (MSF) developed its own solution: the Mini-Lab. The Mini-Lab is a transportable, self-contained, quality-assured, stand-alone CBL adapted to low-resource settings. It can be operated by laboratory technicians without prior microbiology expertise except for a short, one-month training [23,24,25,26].

### 1.2. Development of Adapted AST Solution for LRS within the Context of Mini-Lab

AST techniques have been embarked in the Mini-Lab in order for their users to (1) improve case management of bloodstream infections, (2) support antimicrobial stewardship, and (3) capture data from decentralized rural areas for AMR surveillance [27]. First, technical specifications (target product profile) were defined, a market analysis was performed, and a call for proposal was launched. The MicroScan^®^ (Beckman Coulter, Inc., West Sacramento, CA, USA) platform and the PROMPT^TM^ inoculation system (Beckman Coulter, Inc., West Sacramento, CA, USA) were selected for their ready-to-use and sealed/packaged formats. Moreover, lyophilized MIC AST micro-broth dilution systems were considered less error-prone than disc diffusion methods. They provide first-rate information [28,29], such as MIC, which can be read manually or with an automatic reader, and produces high-reproducibility and standardized results thanks to its pre-prepared panels. MSF partnered with Beckman Coulter to tailor the MIC panel. The selection of antibiotics was based on (i) the list of antibiotics available as CE-IVD from Beckman Coulter, (ii) the list of antibiotics used in MSF facilities, and (iii) the WHO’s essential drug lists [30]. Those panels were tailored to the needs of the patients, the local epidemiology, and expected antibiotic resistance (ABR) patterns (Appendix A). 

Special attention was given to commonly-used antibiotics, antibiotics of last resort, and proxy indicators of resistance mechanisms as per GLASS requirements [27] and AWaRe classifications [31]. Drug dilutions were chosen to match both CLSI and EUCAST breakpoints in 2019 [32,33] and were embedded by Beckman Coulter on the MicroScan panels (Figure 1 and Appendix A); antibiotics abbreviations were defined as per EUCAST recommendations [34].

Three CE-IVD AST microplates were developed: one for rapidly-growing aerobic and facultatively anaerobic gram-positive cocci (MicroScan MSF dried overnight Gram-positive panel, C32698), one for aerobic and facultatively anaerobic gram-negative bacilli (MicroScan MSF dried overnight Gram-negative panel, C32699), and one for aerobic non-enterococcal streptococci (including *Streptococcus pneumoniae* and *Haemophilus* spp.) (MicroScan MSF dried overnight fastidious panel, C32700). 

During the CE-IVD certification process, most of the performance testing by manufacturers relies on isolates from high-income countries, where etiologies of sepsis are usually different from those in LMICs [27,28,29,30]. Therefore, our study aimed (i) to verify the accuracy of the three panels used with the Prompt inoculation methods and isolates either from LRS or challenging strains following ISO20776-2:2007 recommendations [35], and (ii) to verify the inter-observer variability in manually reading the panels

## 2. Materials and Methods

### 2.1. MicroScan MSF MIC Panels

We evaluated the MicroScan MSF dried overnight Gram-positive type 1 (MICPOS1), MicroScan MSF dried overnight Gram-negative type 1 (MICNEG1), and MSF dried overnight fastidious type 1 (MICFAST1) panels. In the current manuscript, we will further use the terms MICPOS1, MICNEG1, and MICFAST1 panels, respectively, to refer to each of these panels. When mentioning all types of panels, we will refer to “MSF MIC panels”. The breakpoints table of EUCAST version 9.1 (2019) [33] was used to interpret the MIC results. The lot numbers of the panels and reagents used in this experiment are listed in Appendix A.

### 2.2. Clinical Isolates and Reference Strains

A total of 387 anonymized clinical isolates, either fresh, recently frozen, or from stock, were tested. These included 332 isolates corresponding to the most common bloodstream pathogens or contaminants in LRS. Of the clinical isolates, 47.4% originated from sub-Saharan Africa, 28.5% from Asia, 12.2% from South America, and 11.4% from Europe (See Table 1 for details per species). On the MICPOS1, 123 Gram-positive strains were tested, of which, 60% (74) were *Staphylococcus* spp. isolates and 40% (49) were *Enterococcus* spp. isolates. On the MICNEG1, 157 Gram-negative rod isolates were tested, of which 72% (112) were Enterobacterales and 28% (45) were non-fermenting Gram-negative rods. On the MICFAST1, 107 fastidious isolates were tested, of which 82% (87) were *Streptococcus* spp. isolates and 18% (20) were *Haemophilus influenzae* isolates. Bacterial isolates were obtained from microbiological surveillance studies in LRS, from the strain collections of the Institute of Tropical Medicine (ITM), Antwerp, Belgium, of the Bicêtre University Hospital, French National Reference Laboratory for Antimicrobial Resistance (French AMR NRL), Paris, France, and of the Hôpital Universitaire Saint-Pierre, Université Libre Bruxelles (LHUB-ULB), Brussels, Belgium. As per ISO recommendations for evaluating the performance of AST [36], at least 25% of the isolates in the entire study were from fresh clinical samples.

### 2.3. Reference Antimicrobial Susceptibility Testing Methods

The antimicrobial susceptibility profiles of the isolates were determined in most cases by disc diffusion (Kirby Bauer) following the EUCAST standard method [37]. Exceptions were: agar gradient diffusion (Liofilchem, Roseto degli Abruzzi, Italy) for teicoplanin and vancomycin with all staphylococci (84), broth microdilution (dried panels from Sensititre, Thermo Fisher Scientific, East Grinstead, UK) for colistin with all Gram-negative bacilli (157) and daptomycin with all staphylococci (84), and agar dilution (Liofilchem, Roseto degli Abruzzi, Italy) for fosfomycin with all staphylococci (84), as per EUCAST guidelines (the complete list of reagents can be found in Appendix A). Reference testing was performed at the same time as MSF panels testing, according to the manufacturer’s instructions, and will be referred to in this article as AST reference panels or reference method. 

### 2.4. Inoculum Preparation

Prior to testing, frozen isolates were subcultured twice and fresh isolates were subcultured once on tryptic soy agar plates containing 5% sheep blood (blood agar plate (BAP)) (or chocolate agar for *H. influenzae*) and incubated at 35 °C for 18 to 24 h under aerobic or CO_2_ atmosphere as per isolate requirement.

For the turbidity methods (MSF MIC Panels), in accordance with the manufacturer instruction for users (IFU), four to five large, or five to ten small well-isolated colonies were collected from an 18–24 h BAP or chocolate agar using a cotton swab and resuspended in 3 mL of Inoculum Water (B1015-2, Beckman Coulter, Inc., West Sacramento CA, USA) for 2–3 s using a vortex. Turbidity was measured using a turbidimeter (Den 1B, Biosan, Riga, Latvia) and adjusted as needed to reach the final turbidity of 0.5 +/− 0.02 McFarland. For the MICPOS1 and MICNEG1, 100 μL (0.1 mL) of the suspension was transferred into a 25 mL tube of Inoculum Water with Pluronic (B1015-7, Beckman Coulter, Inc., West Sacramento CA, USA) and mixed 8–10 times. For the MICFAST1, 100 μL (0.1 mL) of the suspension was transferred into a 25 mL tube of *Haemophilus* Test Medium (HTM) (B1015-26, Beckman Coulter, Inc., West Sacramento CA, USA) for *Haemophilus* spp. isolates or into a 25 mL tube of cation-adjusted Mueller–Hinton Broth with 3% Lysed Horse Blood (LHB) (B1015-25, Beckman Coulter, Inc., West Sacramento CA, USA) for *Streptococcus* spp. isolates and mixed 8–10 times.

### 2.5. Comparison between Standard Inoculum Method and Prompt Inoculation Method

All isolates inoculum-tested with MICPOS1 and MICNEG1 panels were prepared using the turbidity method and the Prompt inoculum method [38]. The Prompt Inoculation System-D (reference B1026-10D, Beckman Coulter, Inc., West Sacramento CA, USA) consists of a rod with a groove at its tip, it is designed to hold a specific number of bacteria (“wand”) and a bottle of diluent for the resuspension of bacteria. A breakaway collar is a small cylinder placed along the wand that serves as a wiping mechanism. Here, the wand was used to touch three different colonies as large as (or larger than) the tip, holding the wand perpendicular to the agar surface, and then the collar was slid down to wipe the wand before placing it into the bottle, pressing down to ensure a tight seal. The bottle was then vigorously shaken 8 to 10 times to resuspend the bacteria from the wand tip. The Prompt microbial suspension was used within 4 h, as stated by IFU. 

### 2.6. Panel Inoculation and Incubation

Purity check plates were performed on all isolates tested using Mueller–Hinton agar or chocolate agar. MSF MIC panels were inoculated using the Renok Rehydrator/Inoculator, a manual pipettor that simultaneously rehydrates and inoculates all 96 wells of MicroScan panels. Contents of the inoculated Pluronic tube used with the turbidity method or of the Prompt bottle were poured into the Seed Tray Renok disposable D-inoculation set (B1013-4, Beckman Coulter, Inc., West Sacramento, CA, USA), an inoculator set consisting of a transfer lid (to hold and dispense the inoculum), and a seed trough (to contain the inoculum). The solution was transferred to the MSF MIC panels using MicroScan Renok (B1018-18, Beckman Coulter), which delivered 115 ± 10 µL of broth suspension to each well. Reference panels were inoculated and incubated according to EUCAST and IFU from the manufacturer. All MSF MIC panels were incubated at 35 +/− 2 °C in an offline, ambient air non-CO_2_ incubator. 

### 2.7. Manual Panel Reading and Inter Observer Variability

The MSF MIC panels were read 16–20 h after incubation. The panels were read manually against a black, indirectly lighted background using a viewer box prototype (Figure 2) adapted for this purpose (Ref. 9999400, JP Selecta, Barcelona, Spain) with interchangeable white or black backgrounds according to the type of panels. Growth in a well was defined as turbidity in the form of haze throughout the well, a button in the center of the well, or fine granular growth throughout the well as per EUCAST [39]. MSF MIC panels were read manually by two technicians. For inter-observer variability calculation, laboratory technicians were blinded to each other’s results. If a discrepancy in reading was found, a consensus was made among readers for the final reading results. All results were recorded onto specific bench sheets and imported into WHONET version 5.6, freely available software for the interpretation of AST using MIC or inhibition zone diameter data [40,41].

### 2.8. Ease of Use 

Assessment of the ease of use was done by surveying the operators with a questionnaire for feedback on each of the components of the system. The readability level of the IFU was assessed using Flesch–Kincaid Grade levels (https://www.online-utility.org/english/readability_test_and_improve.jsp; accessed on 20 March 2021) [13].

### 2.9. Data Analysis

The a priori sample size calculation was not performed before the start of the study. Data were collected on worksheets and entered into Microsoft Excel 2019 (version 2110). The sample size was determined following ISO 20776:2 2007 standard [35] recommendations. Statistical analyses were performed in R (version 4.0.2) using RStudio or Microsoft Excel 2019 (version 2110). The essential agreement (EA) was not calculated as most of the reference testing consisted of disc diffusion methods giving only interpretative category results. Categorical agreement (CA), very major errors (VMEs), major errors (MEs), and minor errors (mEs) were calculated as described in the ISO 20776:2 2007 standard [35]. The CA was defined as susceptible (S), susceptible to increased exposure (I), or resistant (R), as per the EUCAST definition V9, which was the same with both methods. A VME was defined as a false susceptible result with the MSF MIC panels, whereas an ME was a false R or non-susceptible result with the MSF MIC panels; a mE was identified when one method reported an I result while the other method reported S or R results. The acceptance criteria for the study are based on ISO 20776-2:2007 and are as follows: CA ≥ 90%; ME ≤ 3%, VME ≤ 3%.

For inter-observer agreement, two indicators were calculated. First, a measure of the reliability of reading the MIC by a reader against the final reading (an agreement made by both readers if there was a discrepancy) with the calculation of Cohen’s kappa (CK) coefficient [42]. A CK > 0.8 was considered as a very good agreement; 0.6 < CK ≤ 0.8 as a good agreement; 0.4 < CK ≤ 0.6 as a moderate agreement; 0.2 < CK ≤ 0.4 as a fair agreement; and CK ≤ 0.2 as a poor agreement. Second, CA against the AST reference method was calculated by each reader to determine the impact on the result interpretations.

### 2.10. Quality Control

Daily QC was done on the MSF MIC panel tested according to the Beckman recommendations (*E. coli* ATCC1 25922, *P. aeruginosa* ATCC 27853, *S. aureus* ATCC1 29213, *E. faecalis* ATCC 29212, *H. influenzae* ATCC 49766, and *S. agalactiae* ATCC 13813). For out-of-control QC results, after a careful panel examination, QC testing was repeated and if the “out-of-control” occurred again, testing was stopped to identify the problem.

### 2.11. Resolution of Discrepancies

Isolates with a VME or ME were retested using both methods, as were selected isolates with specific drug/organism combinations resulting in ≥10% mEs. Calculations of CA, VMEs, MEs, and mEs were obtained following resolutions of discrepant results after repeated testing. If an error persisted after repeated testing, it was included in the calculations. If the error was resolved after repeated testing, it was not counted as an error, and the initial result was disregarded.

## 3. Results

### 3.1. Evaluation of the MSF MicroScan MIC Panels

#### 3.1.1. MSF Gram-Pos Panel Results

Individual antimicrobial data are presented in Table 2 for *Staphylococcus* spp. and Table 3 for *Enterococcus* spp., CA and error rates were within acceptable limits. Of the 74 *Staphylococcus* spp. tested, after repeated testing, VME occurred in 2 isolates (3%) for teicoplanin and 1 isolate each (1%) for erythromycin and clindamycin. ME occurred in two isolates (3%) for ciprofloxacin, gentamicin, and tetracycline, and one isolate each (1%) for the cefoxitin screening test, amikacin, erythromycin, fosfomycin, trimethoprim/sulfamethoxazole, and tigecycline. Of the 49 *Enterococcus* spp. tested, after repeated testing, 1 (2%) VME was noted for ampicillin, gentamicin high level, and tigecycline. One (2%) ME occurred for ampicillin, ciprofloxacin, gentamicin high level, and tigecycline.

#### 3.1.2. MSF Gram-Neg Panel Results 

Individual antimicrobial data are presented in Table 4 for Enterobacterales and Table 5 for non-fermenting Gram-negative rods. For all isolates tested, CA and error rates were within acceptable limits. Of the 112 Enterobacterales tested, VME occurred in two isolates (3%) for gentamicin and trimethoprim/sulfamethoxazole, and one isolate each (1%) for ampicillin, amoxicillin–clavulanate, amikacin, colistin, and ertapenem. Two isolates (3%) were found to have ME for amoxicillin–clavulanate, ciprofloxacin, and gentamicin, and one (1%) isolate each for ceftazidime, ceftriaxone, trimethoprim/sulfamethoxazole, colistin, and meropenem.

#### 3.1.3. MSF FAST Panel Results

Individual antimicrobial data are presented in Table 6 for Streptococci and Table 7 for *H. influenzae*. For all isolates tested, CA and error rates were within acceptable limits. Of the 87 Streptococci tested, VME was found in one isolate (1%) for vancomycin, clindamycin, trimethoprim/sulfamethoxazole, and chloramphenicol. ME was observed in one (2%) isolate of *Streptococcus pneumoniae* for meropenem and one isolate of *Streptococcus mitis* for penicillin, clindamycin, and trimethoprim/sulfamethoxazole. Of the 20 *Haemophilus influenzae* tested, only 1 was found to be discrepant between I and R (i.e., mE) for trimethoprim/sulfamethoxazole.

Specific resistance mechanism detection data, tested with the MICPOS1 or MICNEG1 panels, are presented in Table 8.

### 3.2. Results of the Evaluation of the Prompt Performance 

Results of the evaluation of MSF MIC panels using the Prompt inoculation method can be found in Appendix A for Staphylococci, Enterococci, Enterobacterales, and non-fermenting gram-negative rods, respectively. Specific resistance mechanism detection data are presented in Appendix A. When Staphylococci were tested with Prompt, the amikacin molecule had the largest number of discrepancies with a CA of 78%, of which 11 and 5 isolates were found to have mE and ME, respectively. Compared with the turbidity inoculum method, more ME in Staphylococci were observed for ciprofloxacin (9%), gentamicin (9%), vancomycin (4%), erythromycin (9%), trimethoprim/sulfamethoxazole (14%), and linezolid (5%), all above the 3% threshold of ISO 20776-2: 2007. For Enterococci, four (9%) isolates had mEs for quinupristin–dalfopristin and two (4%) isolates had VMEs for tigecycline.

### 3.3. Inter-Observer Variability

The results of the inter-observer agreement between the two independent readers who read the panels visually are presented in Table 9. Overall, the reading agreements for MICNEG1 and MICFAST1 were both classified as very good (CK 0.94 and 0.95, respectively). The reading agreement of MICPOS1 was classified as good (CK 0.82). This was mainly due to the lower agreement in reading for ampicillin (0.76), amikacin (0.78), teicoplanin (0.79), and tigecycline (0.82), and especially for daptomycin (0.37), fosfomycin (0.63), and linezolid (0.37). However, the discrepancies in MIC reading did not affect the CA between each of the two readers and the reference method, with CAs of 96% and 96% for daptomycin, 76% and 77% for fosfomycin, and 97% and 96% for linezolid.

### 3.4. Ease of Use

The instructions for interpretation of growth results were considered by the laboratory technicians as understandable but, as no pictures were included to ease the comprehension, the color atlas document made by EUCAST was much appreciated [39]. The Flesch–Kincaid grade levels (FKGL) of the three MSF MIC panels IFU were nine each. FKGL refers to US grade levels (i.e., years of schooling) necessary to understand the text. The FKGL of the first part of the IFU of the Prompt method, with all instructions and limitations (Figure 3a), was 11. The FKGL dropped to six in the last part, where instructions for the Prompt are explained as bench aids with black and white figures (Figure 3b). The FKGL of the IFU of the Renok, which included bench aids as well, was rated 6.

Both the Prompt™ and the RENOK systems were considered user-friendly and time-efficient by both users, particularly compared to other inoculation methods using 0.5 McFarland standards and single-pipette dispensers. Interpreting the panels was not considered difficult, except for trimethoprim/sulfamethoxazole because of the “trailing effect”, a typical reading of this antibiotic where the MIC should be read as: (1) approximately 80% reduction of growth, (2) a white button less than 2 mm in diameter, or (3) a white button that is semi-translucent.

The packaging of individual panels was of very good quality, providing air-sealed individual aluminum–plastic pouches with humidity indicators; they were easy to open. Other components were provided within sturdy cardboard boxes fit for difficult transport conditions in LRS. Concerning the shelf life of the MSF MIC panels, Prompt and Pluronic water were within the limits predefined by the Mini-Lab target product profile (minimally 12 months), and the storage conditions (2–25 °C) were within the acceptable Mini-Lab specifications (2–40 °C). However, the shelf life of the HTM and LHB broth needed to rehydrate MICFAST1 was 6 months at 2–8 °C. 

## 4. Discussion

### 4.1. Performance Evaluation of the MSF MIC Panels

For all drug/organism combinations, our study showed that MICPOS1, MICNEG1, and MICFAST1 panels performed satisfactorily, in agreement with the previous evaluation using the MicroScan technology [43,44,45,46,47], and when testing isolates from LMICs. In addition, they performed as expected with challenging strains for confirmation of resistance mechanisms, such as extended-spectrum beta-lactamase-producing Enterobacterales, methicillin-resistant *Staphylococcus aureus*, induced resistance to clindamycin among Staphylococci, high-level aminoglycoside resistance among Enterococci, and screening of carbapenemase-producing Enterobacterales, *Acinetobacter baumannii*, and *Pseudomonas aeruginosa*. As mentioned by the manufacturer, a limitation of the MICFAST1 panel is its inability to detect resistance to levofloxacin in *Streptococcus* spp. and resistance to ciprofloxacin and meropenem in *Haemophilus influenzae*, due to the lack of resistant strains at the time of comparative testing.

However, before repeated testing for the resolution of discrepant results, amoxicillin–clavulanate (AMC) of the MICNEG1 was found to have a CA of 89% with 11% ME. All MEs (12/112) on the AMC were found in *Salmonella* species, 5 *Salmonella enterica* serotype Choleraesuis (from Cambodia, Ecuador), and 7 *S.* Typhimurium (from DRC, Burkina Faso). When these strains were removed from the analysis, the CA for AMC was 100%. We suspected that the presence of monoclonal heteroresistance was not captured by the disc diffusion methods we used as reference. This has previously been described for other organisms/drug combinations [48,49,50] and colistin, polymyxin, and carbapenems among *S.* Typhimurium [51]. Therefore, we re-evaluated the 12 discrepant isolates using various combinations of discs and media from different manufacturers in triplicate with the addition of another broth microdilution (BMD) panel (Sensititre, Thermo Fisher Scientific, East Grinstead, UK) and gradient diffusion strip (E-test, bioMérieux, Marcy-l’ Étoile, France). The results are presented in Appendix A and photos of the inhibition zone produced by Salmonella Typhimurium are presented in Figure 3. Sensititre BMD and MICNEG1 gave similar MIC results for AMC. Only the combination of BioMérieux specific Mueller–Hinton agar for Enterobacterales (bioMérieux, Marcy-l’ Étoile, France) with a disc from i2a (Montpellier, France) gave similar results to the MICNEG1. To our knowledge, heteroresistance for amoxicillin–clavulanate within the *Salmonella* species has not been described in the literature and should be further studied using a population analysis profile and other methods, as proposed by Andersson et al. in their recent review on the impact of heteroresistance [52].

### 4.2. Performance Evaluation of the Prompt Inoculation Method

The manufacturer IFU stated that the Prompt System demonstrated elevated MICs with fluoroquinolones (e.g., gatifloxacin), lincosamides (e.g., clindamycin), and macrolides (e.g., erythromycin), and a potential ME with tigecycline and Staphylococci, when compared with the reference method, for reasons that were not able to be identified from the literature. Our experiment showed a similar increase in MIC when using the Prompt inoculation method with ciprofloxacin and erythromycin, which impacted the ME rates for Staphylococci coming from LRS but we did not find an increase in the ME rate with tigecycline and clindamycin. Moreover, our results suggest that the Prompt inoculation methods increased MIC and MEs on aminoglycosides, such as gentamicin, amikacin, vancomycin, trimethoprim–sulfamethoxazole, and linezolid for Staphylococci

Other studies suggest and confirm that the Prompt inoculation method has no effect on the MIC for the aerobic and facultatively anaerobic Gram-negative bacilli and Enterococci when compared to the standard inoculum method [38,53].

### 4.3. Inter-Observer Variability

Overall, the agreements between readers were very good, with four different readers during the entire duration of the experiment. No studies reporting agreements between visual readers using Dried overnight MicroScan panels have been published. Despite the first impression of reading difficulties and some disagreements in the reading of some antibiotic MICs, reader discrepancies had no impact on the final clinical category result interpretations.

### 4.4. Adaptation to LRS: Stability, Ease-of-Use 

The temperature stability of the MicroScan panels, currently assured up to 25 °C, does not entirely fulfill the requirements for tropical settings because cool storage (<30 °C) is not always feasible [54]; however, when compared to disc diffusion that requires cold chain storage of the disc, it is more achievable for the MSF supply system in a district hospital to have access to storage facilities with air conditioning than shipping and storing in temperatures between 2 and 8 °C. Regarding the ease-of-use, the MicroScan inoculation system and panel reading (Renok and Prompt for Gram-negative bacilli) is positive; it does not require multiple steps to inoculate panels and the interpretation of panels is quite easy, with the exception of the trimethoprim–sulfamethoxazole (TRS) wells that require some practice. The dedicated prototype microplate viewer box greatly facilitated the reading process. The Flesch–Kincaid grade level scores of the IFU suggested that for MICPOS1, MSFNEG1, and MSFFAST1, a fair level of schooling is required to understand the IFU. Although the language used in professional documents may be at a slightly higher level, Flesch–Kincaid levels below six are desirable for IFU [55,56] as well as for the bench aid parts of the Prompt and Renok systems.

### 4.5. Recommendations for Use and Further Development of the MicroScan System

The current limitations of the Prompt and IFU panels explained above were tackled within the Mini-Lab project, by developing training material, such as videos, adapted laboratory procedures, bench aids, a color atlas of the different types of growth, and a microplate viewer for visual reading (see Appendix A). This evaluation allowed us to adapt our recommendations to field workers, as, for example, to avoid the Prompt when suspecting Staphylococci. In addition, we encourage the manufacturer to mitigate (to the best possible extent) the issues described above. Bench aids could be included with the product as well as video training (available on their website) showing the different growth types from the wells. Lastly, extended shelf-life testing and stability testing in tropical environments are necessary to assure product quality in LRS.

### 4.6. Strengths and Limitations of the Study 

To our knowledge, this is the first study that evaluated the MicroScan Dried MICs on clinical isolates with typical LRS pathogens [57,58]. Furthermore, we assessed robustness and ease of use. We have several limitations to the verification of some drug/organism combinations as we were lacking a number of resistant isolates; amikacin, teicoplanin, vancomycin, quinupristin–dalfopristin, clindamycin, daptomycin, linezolid, tigecycline for Staphylococci; fosfomycin, tigecycline for Enterobacterales; amikacin, piperacillin-tazobactam, chloramphenicol, and colistin for non-fermenting Gram-negative bacilli. Because we did not dispose of the Biosafety Level-3 (BSL-3) facilities, we could not test pathogens such as *Burkholderia pseudomallei*. No inter-user, intra-lot, or intra-method repetition was done. Furthermore, comparing disc diffusion to MIC values is by itself a limitation of this study, but would have not been financially possible.

## 5. Conclusions

Confronted with clinical isolates from LRS, MicroScan dried overnight MIC tailored for MSF had excellent performance for Gram-negative, Gram-positive, and fastidious organisms. The Prompt inoculation system together with the Renok transfer system is very convenient but cannot be used for *Staphylococci*. The study additionally identified potential improvements in stability, robustness, and ease of use to ensure adaptation of the MicroScan system to the constraints of LRS for use outside of the MSF Mini-Lab setting and highlight underseen heteroresistance with the disc diffusion method used to test amoxicillin–clavulanate with Salmonella species, which should be further studied.

## Figures and Tables

**Figure 1 diagnostics-12-02106-f001:**
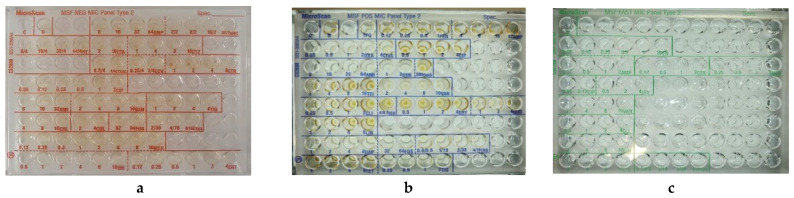
Example of the three MSF MicroScan MIC panels. (**a**) MicroScan MSF dried overnight Gram-positive panel, (**b**) MicroScan MSF dried overnight Gram-negative panel, (**c**) MSF dried overnight fastidious panel.

**Figure 2 diagnostics-12-02106-f002:**
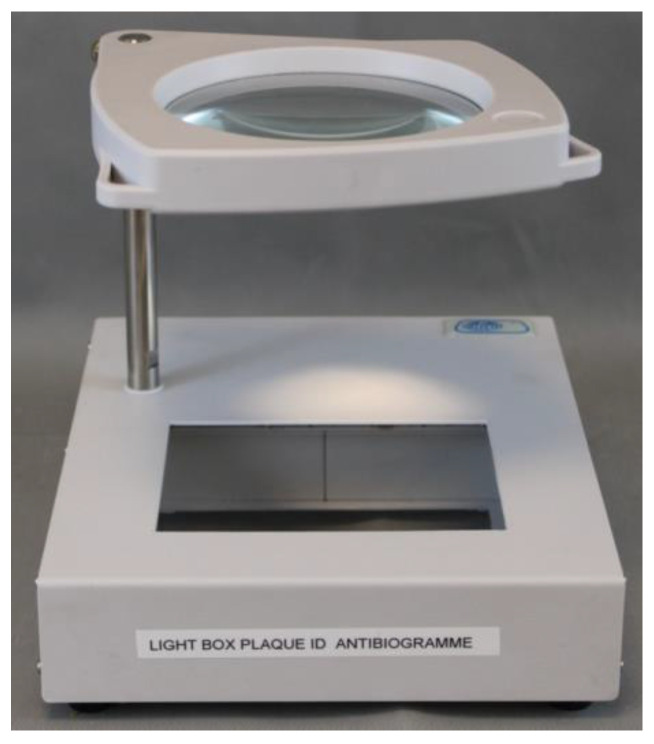
Prototype of the microplate viewer box by JP Selecta, used for visual reading. The background at the bottom can be changed from black to white [26].

**Figure 3 diagnostics-12-02106-f003:**
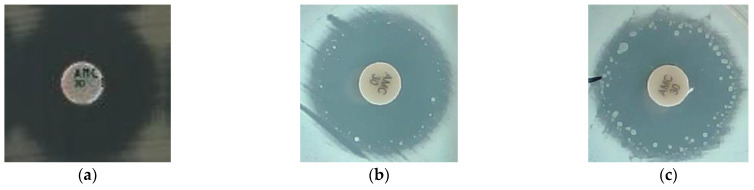
Photo of the inhibition zone produced by *Salmonella* Typhimurium with: (**a**) initial reference method with Mueller–Hinton agar from Becton, Dickinson and a disc from Bio-Rad with no sign of heteroresistance; (**b**,**c**) sign of heteroresistance when using bioMérieux Mueller–Hinton Enterobacterales agar (bioMérieux Inc., Marcy l’ Étoile, France) and an AMC disc from i2a (i2a, Montpellier, France).

**Table 1 diagnostics-12-02106-t001:** Geographical origins of the isolates. Anonymized isolates were obtained from surveillance studies of several partners (ITM, French AMR NRL, LHUB-ULB).

Species	Total Number of Isolates Tested	Africa	Asia	South America	Europe
Species Tested on the MSF Pos MIC Panel					
*Staphylococcus aureus*	47	23	13	6	5
*Staphylococcus epidermidis*	11	5	3	1	2
*Staphylococcus hominis*	15	7	4	2	2
*Staphylococcus haemolyticus*	10	5	3	1	1
*Staphylococcus warneri*	1	1	-	-	-
*Enterococcus faecium*	35	14	10	4	4
*Enterococcus faecalis*	14	6	4	2	2
Total isolates tested on the panel	133	61	37	16	16
Species Tested on the MSF Neg MIC Panel					
*Escherichia coli*	25	12	7	3	3
*Klebsiella pneumoniae*	29	14	8	4	3
*Klebsiella oxytoca*	9	4	3	1	1
*Klebsiella ozaena*	1	1	-	-	-
*Morganella morganii*	1	1	-	-	-
*Salmonella* Paratyphi A	8	4	2	1	1
*Salmonella* Typhimurium ^a^	8	4	2	1	1
*Salmonella* Choleraesuis	7	3	2	1	1
*Enterobacter cloacae*	16	7	5	2	2
*Enterobacter hermannii*	1	1	-	-	-
*Enterobacter kobei*	1	1	-	-	-
*Enterobacter asburiae*	1	1	-	-	-
*Citrobacter freundii* complex	5	2	1	1	1
*Pseudomonas aeruginosa*	15	7	4	2	2
*Acinetobacter baumannii complex*	14	6	4	2	2
*Burkholderia cepacia*	10	5	3	1	1
*Stenotrophomonas maltophilia*	6	3	2	1	-
Total isolates tested on the panel	157	76	43	20	18
Species Tested on the MSF FAST MIC Panel					
*Streptococcus pneumoniae*	54	26	15	7	6
*Streptococcus agalactiae*	37	17	12	4	4
*Streptococcus pyogenes*	30	14	9	4	3
*Streptococcus mitis*	22	10	6	3	2
*Streptocococcus oralis*	6	3	2	1	1
*Streptococcus anginosus*	10	5	3	1	1
*Streptococcus constellatus*	2	1	1	-	-
*Haemophilus influenzae*	20	10	6	2	2
Total isolates tested on the panel	181	86	54	22	19

^a^*S. enterica* subsp. enterica serovar Typhimurium (hereafter, *S.* Typhimurium).

**Table 2 diagnostics-12-02106-t002:** Results for Staphylococci tested with the MICPOS1 standard turbidity inoculum method and visual reading versus AST reference methods.

	No. of Isolates *AST Reference* ^a^				No. of Isolates MICPOS1 ^b^						
Antimicrobial	Total	R	I	S	R	I	S	CA ^c^ (no. [%])	mE ^d^ (no. [%])	ME ^e^ (no. [%])	VME ^f^ (no. [%])
Penicillin	63	59	0	4	59	0	4	63 (100)	0 (0)	0 (0)	0 (0)
Ciprofloxacin	74	37	0	37	39	0	35	72 (97)	0 (0)	2 (3)	0 (0)
Amikacin	74	17	4	53	16	8	50	69 (93)	4 (5)	1 (1)	0 (0)
Gentamicin	74	28	0	46	30	0	44	72 (97)	0 (0)	2 (3)	0 (0)
Teicoplanin	74	3	0	71	1	0	73	72 (97)	0 (0)	0 (0)	2 (3)
Vancomycin	74	2	0	72	2	0	72	74 (100)	0 (0)	0 (0)	0 (0)
Quinupristin-dalfopristin	74	0	7	67	0	7	67	74 (100)	0 (0)	0 (0)	0 (0)
Erythromycin	74	30	0	44	30	1	43	71 (96)	1 (1)	1 (1)	1 (1)
Clindamycin	74	14	0	60	12	1	61	72 (97)	1 (1)	0 (0)	1 (1)
Daptomycin	74	1	0	73	1	0	73	74 (100)	0 (0)	0 (0)	0 (0)
Fosfomycin	74	29	0	45	30	0	44	73 (99)	0 (0)	1 (1)	0 (0)
Trimethoprim/Sulfamethoxazole	74	18	0	56	19	3	52	70 (95)	3 (4)	1 (1)	0 (0)
Linezolid	74	5	0	69	5	0	69	74 (100)	0 (0)	0 (0)	0 (0)
Tetracycline	74	37	0	37	39	0	35	72 (97)	0 (0)	2 (3)	0 (0)
Tigecycline	74	0	0	74	1	0	73	73 (99)	0 (0)	1 (1)	0 (0)

^a^ Number of isolates tested with the reference method and classified as R resistant; I, susceptible, increased exposure; S, susceptible; ^b^ number of isolates tested with the evaluated method and classified as R resistant; I, susceptible, increased exposure; S, susceptible; ^c^ CA, categorical agreement; ^d^ mE, minor error; ^e^ ME, major error; ^f^ VME, very major error.

**Table 3 diagnostics-12-02106-t003:** Results for Enterococci tested with the MICPOS1 standard turbidity inoculum method and visual reading versus AST reference methods.

	No. of Isolates *AST Reference* *^a^*				No. of Isolates MICPOS1 *^b^*						
Antimicrobial	Total	R	I	S	R	I	S	CA *^c^* (no. [%])	mE *^d^* (no. [%])	ME ^e^ (no. [%])	VME *^f^* (no. [%])
Ampicillin	49	30	0	19	29	1	18	47 (96)	0 (0)	1 (2)	1 (2)
Ciprofloxacin	49	29	0	20	30	0	19	48 (98)	0 (0)	1 (2)	0 (0)
Teicoplanin	49	23	0	26	23	0	26	49 (100)	0 (0)	0 (0)	0 (0)
Vancomycin	49	23	0	26	23	0	26	49 (100)	0 (0)	0(0)	0 (0)
Quinupristin-dalfopristin	49	19	16	14	19	16	14	45 (91)	4 (9)	0 (0)	0 (0)
Linezolid	49	2	0	47	2	0	47	49 (100)	0 (0)	0 (0)	0 (0)
Tigecycline	49	4	0	45	4	0	45	47 (96)	0 (0)	1 (2)	1 (2)

^a^ Number of isolates tested with the reference method and classified as R resistant; I, intermediate; S, susceptible; ^b^ number of isolates tested with the evaluated method and classified as R resistant; I, intermediate; S, susceptible; ^c^ CA, categorical agreement; ^d^ mE, minor error; ^e^ ME, major error; ^f^ VME, very major error.

**Table 4 diagnostics-12-02106-t004:** Results for Enterobacterales tested with the MICNEG1 standard turbidity inoculum method and visual reading versus AST reference methods.

	No. of Isolates *AST Reference* *^a^*				No. of Isolates MICPNEG1 *^b^*						
Antimicrobial	Total	R	I	S	R	I	S	CA *^c^* (no. [%])	mE *^d^* (no. [%])	ME ^e^ (no. [%])	VME *^f^* (no. [%])
Ampicillin	112	100	0	12	99	0	13	111 (99)	0 (0)	0 (0)	1 (1)
Amoxicillin/Clavulanic Acid	112	72	0	40	73	0	39	109 (97)	0 (0)	2 (2)	1 (1)
Ceftazidime	112	47	2	63	49	0	64	108 (96)	3 (3)	1 (1)	0 (0)
Ceftriaxone	112	50	0	62	51	1	60	110 (98)	1 (1)	1 (1)	0 (0)
Piperacillin/Tazobactam	112	39	4	69	39	4	69	106 (93)	6 (7)	0 (0)	0 (0)
Ciprofloxacin	112	51	4	57	53	4	55	106 (93)	4 (4)	2 (2)	0 (0)
Amikacin	112	16	4	92	14	5	93	106 (93)	5 (6)	0 (0)	1 (1)
Gentamicin	112	40	5	67	40	4	68	104 (92)	4 (4)	2 (2)	2 (2)
Trimethoprim/Sulfamethoxazole	112	69	1	42	69	0	43	108 (96)	1 (1)	1 (1)	2 (2)
Chloramphenicol	112	47	0	65	47	0	65	112 (100)	0 (0)	0 (0)	0 (0)
Colistin	112	12	0	77	12	0	77	110 (98)	0 (0)	1 (1)	1 (1)
Fosfomycin	112	1	0	111	1	0	111	112 (100)	0 (0)	0 (0)	0 (0)
Tigecycline	112	0	0	112	0	0	112	112 (100)	0 (0)	0 (0)	0 (0)
Meropenem	112	16	6	90	17	6	79	106 (93)	5 (6)	1 (1)	0 (0)
Imipenem	112	19	7	86	19	7	86	112 (100)	0 (0)	0 (0)	0 (0)
Ertapenem	112	33	0	79	32	0	80	111 (99)	0 (0)	0 (0)	1 (1)

^a^ Number of isolates tested with the reference method and classified as R resistant; I, susceptible, increased exposure; S, susceptible; ^b^ number of isolates tested with the evaluated method and classified as R resistant; I, susceptible, increased exposure; S, susceptible; ^c^ CA, categorical agreement; ^d^ mE, minor error; ^e^ ME, major error; ^f^ VME, very major error.

**Table 5 diagnostics-12-02106-t005:** Results for non-fermenting Gram-neg bacilli tested with the MICNEG1 standard turbidity inoculum method and visual reading versus AST reference methods.

	No. of Isolates *AST Reference* *^a^*				No. of Isolates MICPNEG1 *^b^*						
Antimicrobial	Total	R	I	S	R	I	S	CA *^c^* (no. [%])	mE *^d^* (no. [%])	ME ^e^ (no. [%])	VME *^f^* (no. [%])
Ceftazidime	15	8	3	4	8	2	5	14 (96)	1 (4)	0 (0)	0 (0)
Piperacillin/Tazobactam	15	4	0	11	4	0	11	15 (100)	0 (0)	0 (0)	0 (0)
Ciprofloxacin	29	12	6	11	12	6	11	29 (100)	0 (0)	0 (0)	0 (0)
Amikacin	29	7	2	20	7	2	20	29 (100)	0 (0)	0 (0)	0 (0)
Gentamicin	29	15	0	14	15	0	14	29 (100)	0 (0)	0 (0)	0 (0)
Trimethoprim/Sulfamethoxazole	29	10	0	20	9	1	20	28 (99)	1 (1)	0 (0)	0 (0)
Chloramphenicol	9	4	3	2	4	3	2	9 (100)	0 (0)	0 (0)	0 (0)
Colistin	15	3	0	12	3	0	12	15 (100)	0 (0)	0 (0)	0 (0)
Meropenem	45	14	6	25	14	6	25	43 (96)	2 (4)	0 (0)	0 (0)
Imipenem	29	12	0	17	10	1	18	27 (94)	1 (3)	0 (0)	1 (3)

^a^ Number of isolates tested with the reference method and classified as R resistant; I, susceptible, increased exposure; S, susceptible; ^b^ number of isolates tested with the evaluated method and classified as R resistant; I, susceptible, increased exposure; S, susceptible; ^c^ CA, categorical agreement; ^d^ mE, minor error; ^e^ ME, major error; ^f^ VME, very major error.

**Table 6 diagnostics-12-02106-t006:** Results for *Streptococcus* spp. test tested with the MICFAST1 standard turbidity inoculum method and visual reading versus AST reference methods.

	No. of Isolates *AST Reference* *^a^*				No. of Isolates MICPFAST1 *^b^*						
Antimicrobial	Total	R	I	S	R	I	S	CA *^c^* (no. [%])	mE *^d^* (no. [%])	ME ^e^ (no. [%])	VME *^f^* (no. [%])
Penicillin	87	10	9	68	12	8	67	84 (97)	2 (2)	1 (1)	0 (0)
Meropenem _g_	53	0	0	53	1	0	52	52 (98)	0 (0)	1 (2)	0 (0)
Ceftriaxone _f_	-	-	-	-	-	-	-	-	-	-	-
Ampicillin _g_	53	12	6	35	14	2	37	49 (92)	4 (8)	0 (0)	0 (0)
Levofloxacin _h_	63	1	0	62	1	0	62	63 (100)	0 (0)	0 (0)	0 (0)
Vancomycin	87	2	0	85	1	0	86	86 (99)	0 (0)	0 (0)	1 (1)
Clindamycin	87	16	0	71	16	0	71	85 (98)	0 (0)	1 (1)	1 (1)
Trimethoprim/Sulfamethoxazole	87	4	0	83	4	0	83	85 (98)	0 (0)	1 (1)	1 (1)
Chloramphenicol	87	3	0	84	2	0	85	86 (99)	0 (0)	0 (0)	1 (1)
Linezolid _i_	63	0	0	63	0	0	63	63 (100)	0 (0)	0 (0)	0 (0)

^a^ Number of isolates tested with the reference method and classified as R resistant; I, susceptible, increased exposure; S, susceptible; ^b^ number of isolates tested with the evaluated method and classified as R resistant; I, susceptible, increased exposure; S, susceptible; ^c^ CA, categorical agreement; ^d^ mE, minor error; ^e^ ME, major error; ^f^ VME, very major error; ^g^ The susceptibility of streptococcus group A, B, C, G to cephalosporins is inferred from the benzylpenicillin susceptibility, no breakpoint available on disc diffusion; ^h^ Only interpretation for *S. pneumoniae* and *S. viridans* group; ^i^ Only interpretation for *S. pneumoniae*, *S. viridans* group, and *S. anginosus* group.

**Table 7 diagnostics-12-02106-t007:** Results for the *H. influenzae* test with the MICFAST1 standard turbidity inoculum method and visual reading versus AST reference methods.

	No. of Isolates *AST Reference* *^a^*				No. of Isolates MICPFAST1 *^b^*						
Antimicrobial	Total	R	I	S	R	I	S	CA *^c^* (no. [%])	mE *^d^* (no. [%])	ME ^e^ (no. [%])	VME *^f^* (no. [%])
Meropenem	20	0	0	20	0	0	20	20 (100)	0 (0)	0 (0)	0 (0)
Ceftriaxone	20	3	0	17	3	0	17	20 (100)	0 (0)	0 (0)	0 (0)
Ampicillin	20	2	0	18	2	0	18	20 (100)	0 (0)	0 (0)	0 (0)
Ciprofloxacin	20	3	0	17	3	0	17	20 (100)	0 (0)	0 (0)	0 (0)
Levofloxacin	20	2	0	18	2	0	18	20 (100)	0 (0)	0 (0)	0 (0)
Trimethoprim/Sulfamethoxazole	20	6	1	13	7	0	13	19 (95)	1 (5)	0 (0)	0 (0)
Chloramphenicol	20	0	0	20	0	0	20	20 (100)	0 (0)	0 (0)	0 (0)

^a^ Number of isolates tested with the reference method and classified as R resistant; I, susceptible, increased exposure; S, susceptible; ^b^ number of isolates tested with the evaluated method and classified as R resistant; I, susceptible, increased exposure; S, susceptible; ^c^ CA, categorical agreement; ^d^ mE, minor error; ^e^ ME, major error; ^f^ VME, very major error.

**Table 8 diagnostics-12-02106-t008:** Results for specific resistance tests using MICNEG1 or MICPOS1 standard turbidity inoculum methods and visual reading versus AST reference methods.

No. of Isolates	*AST Reference* * ^a^ *			MSF Panel *^b^*				
Multidrug Resistant Organism	Total	R	S	R	S	CA *^c^* (no. [%])	ME ^d^ (no. [%])	VME *^e^* (no. [%])
Methicillin-resistant *Staphylococcus aureus* *_f_*	47	33	14	33	14	47 (100)	0 (0)	0 (0)
Inducible clindamycin-resistant Staphylococci *_f_*	74	26	48	26	48	74 (100)	0(0)	0 (0)
Vancomycin-resistant *Staphylococcus aureus* *_f_*	47	2	45	2	45	47 (100)	0 (0)	0 (0)
High-level gentamicin resistance Staphylococci *_f_*	49	25	24	25	24	47 (96)	1 (2)	1 (2)
Vancomycin-resistant Enterococci *_f_*	49	23	26	23	26	49 (100)	0(0)	0 (0)
Extended-spectrum beta-lactamase-producing Enterobacterales *_g_*	112	19	93	19	93	112 (100)	0 (0)	0 (0)
Carbapenem-resistant Enterobacterales *_g_*	112	33	79	32	80	111 (99)	0 (0)	1 (1)
Colistin-resistant Enterobacterales *_g_*	112	12	77	12	77	110 (98)	1 (1)	1 (1)
Carbapenem-resistant *Pseudomonas aeruginosa* *_g_*	15	4	11	4	11	15 (100)	0 (0)	0 (0)
Colistin-resistant *Pseudomonas aeruginosa* *_g_*	15	4	11	4	11	15 (100)	0 (0)	0 (0)
Carbapenem-resistant *Acinetobacter baumannii* *_g_*	14	8	6	8	6	15 (100)	0 (0)	0 (0)

^a^ Number of isolates tested with the reference method and classified as R resistant; I, susceptible, increased exposure; S, susceptible; ^b^ number of isolates tested with the evaluated method and classified as R resistant; I, susceptible, increased exposure; S, susceptible; ^c^ CA, categorical agreement; ^d^ ME, major error; ^e^ VME, very major error; ^f^ Resistance test evaluated on the MICPOS1 panel; ^g^ Resistance test evaluated on the MICNEG1 panel.

**Table 9 diagnostics-12-02106-t009:** Results of the inter-observer variability test using MSF MIC panels with the Prompt inoculation method versus the standard AST method.

	*Staphylococci and Enterococci*				*Gram-negative bacilli*				*Fastidious organisms*			
Antimicrobial	n ^a^	Kappa Cohen	R1 CA ^b^ [%]	R2 CA _c_ [%]	n ^a^	Kappa Cohen	R1 CA ^b^ [%]	R2 CA ^c^ [%]	n ^a^	Kappa Cohen	R1 CA ^b^ [%]	R2 CA ^c^ [%]
Penicillin	63	0.95	98	98					87	0.99	97	97
Ampicillin	123	0.76	96	92	112	0.91	99	98	107	1.00	98	98
Amoxicillin/Clavulanic Ac					112	0.97	99	98				
Cefoxitin screening/Oxacillin	74	0.95	92	93								
Ceftazidime					127	0.95	97	96				
Ceftriaxone					112	0.89	97	98	20	1.00	100	100
ESBL test					112	0.97	99	97				
Piperacillin/Tazobactam					127	0.97	99	93				
Levofloxacin									83	0.98	100	100
Ciprofloxacin	49	0.89	99	99	141	0.95	94	96				
Amikacin	123	0.78	78	76	141	0.99	99	93				
Gentamicin	123	0.92	100	96	141	1.00	92	92				
Gentamicin (high level)	49	0.90	90	90								
Teicoplanin	123	0.79	99	96								
Vancomycin	123	1.00	97	97					87	1.00	98	98
Quinupristin-dalfopristin	123	0.93	89	91								
Erythromycin	74	0.97	96	96								
Clindamycin	74	0.99	97	97					87	0.64	100	100
Inducible clindamycin resistance	74	0.96	97	95								
Daptomycin	74	0.37	96	96								
Fosfomycin	74	0.63	76	77	112	0.76	100	100				
Trimethoprim/Sulfamethoxazole	74	0.62	95	95	142	0.87	96	96	107	0.92	98	93
Linezolid	123	0.37	97	96					63	1.00	100	100
Tetracycline	74	0.94	97	97								
Tigecycline	123	0.82	95	94	112	0.97	100	100				
Chloramphenicol					121	0.90	99	99	107	1.00	98	98
Colistin					127	0.97	98	98				
Meropenem					157	0.96	93	93	73	0.97	98	98
Imipenem					141	0.96	100	100				
Ertapenem					112	0.97	99	99				
Average all molecules		0.82	94	93		0.94	98	97		0.95	99	98

^a^ Total number of isolates tested by the molecule. ^b^ R1CA categorical agreement between Reader 1 and reference methods. ^c^ R2CA categorical agreement between Reader 1 and reference methods.

## Data Availability

The data will be available upon reasonable request from the corresponding author.

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
