# Peer review of "Validation of Three MicroScan® Antimicrobial Susceptibility Testing Plates Designed for Low-Resource Settings"

_diagnostics, 2022, doi:10.3390/diagnostics12092106_

Round 1
Reviewer 1 Report
This is an interesting article that compares standard AST techniques such as disc diffusion to newly developed MicroScan plates. The use of these plates would be highly advantageous for low- and middle- income settings with limited human capacity and training.
Comments-
Overall the English needs to be improved. Please use words instead of numbers if number is <10 and is not followed by a unit of measurement.
Line 167-172- please check English.
Line 239- Staphylococcus spp. not Staphylococcus. spp
Materials and methods
Please include further details on- specimen types, sampling periods for the isolates, the countries isolates are from and preferably the organisations, and what QC was carried out (eg. control strains used on the plates)
Line 138- 146- please state the number of isolates which were tested by each method
Line 150- please include the incubation atmosphere
Line 197- please state version of WHONET
Section 2.9 Please add what programs were used for statistical analysis
Results
Line 241- by molecules do you mean antibiotic?
Table 2-7- please give the results for all the standard turbidity method and AST reference methods in the table so the reader can see where the differences were
Lines 345- 353 are repeated in lines 359-367
Table 3- I and S should not be italics
Supplementary table S4-S7- please include full details similar to suggestion above for tables 2-7
Discussion
In the discussion you mention how good the test is at detecting ESBLs, CREs and MRSA, but there is no mention of these isolates in the methods or the results. Were they tested in this study and what were the results?
What is the cost of the plates compared to disc diffusion?
Limitations section- another limitation is that you are comparing disc diffusion to MIC values. It is understandable that setting up E-tests or other MIC methods probably would not have been financially possible in these settings for comparison, but should just be mentioned in the limitations
Supplementary file
Table S8- for isolate 135 results for BMD MIC-NEG1 and BMD Sensi-titre look wrong
Author Response
Dear Editor,
We would sincerely like to thank the reviewers for reading our manuscript and providing their valuable comments. In this letter, we will try to address some of their concerns. We hope we will be able to answer them in a satisfactory manner.
Reviewer comments# 1:
Point 1: Overall, the English needs to be improved. Please use words instead of numbers if number is <10 and is not followed by a unit of measurement.
Response 1: Thank you for this comment, English has been revised by a native English speaker and changes to numbers were made accordingly.
Point 2: Line 167-172- please check English.
Response 2: We have rephrased entirely the paragraph to make sure it is more understandable for readers
Point 3: Line 239- Staphylococcus spp. not Staphylococcus. Spp
Response 3: Thank you for identifying this inconsistency, changes were made accordingly
Materials and methods
Point 4: Please include further details on- specimen types, sampling periods for the isolates, the countries isolates are from and preferably the organizations, and what QC was carried out (eg. control strains used on the plates)
Response 4: Thank you for this pertinent comment, all Isolate received from partners where clinical anonymized isolate, therefore we are not able to have much information than the fact that they are coming from patients with bloodstream infection and the country of origin, we have added line 188 the fact that they are “clinical anonymized isolates”. We stated in the Institutional Review Board Statement section that bacterial isolates were obtained from microbiological surveillance studies in LRS (surveillance of antimicrobial resistance among consecutive blood culture isolates in tropical settings, Institutional Review Board number 613/08, Ethical Committee Antwerp University Hospital 8/20/96) or from anonymous strain collections of Institute of Tropical Medicine (ITM) travel clinic, Antwerp, Belgium or the Bicêtre University Hospital, French National Referral Laboratory for Antibiotic Resistance (French AMR NRL) and of the Saint Pierre Laboratoire Hospitalo-Universitaire Brussel-Université Libre Bruxelle (LHUB-ULB) and were not authorized to retrieve further information. For quality control we have added the information on the control that were performed
Point5: Line 138- 146- please state the number of isolates which were tested by each method the same 356 isolate where tested with the same method
Response 5: We have included in this paragraph for the method outside of the Disc diffusion, the antibiotic tested for what species by what method and the number of isolate tested.
Point 6: Line 150- please include the incubation atmosphere
Response 6: We have added the following sentence “under aerobic or C02 atmosphere as per isolate requirement”
Point 7: Line 197- please state version of WHONET
Response 7: At the time of the experiment, the WHONET 5.6 version has been used, we have included the version number
Point 8: Section 2.9 Please add what programs were used for statistical analysis
Response 8: In the section mentioned we have added the following sentence “Data was collected on worksheets and entered into Microsoft Excel 2019 (version 2110). Statistical analyses were done in R using RStudio (version 4.0.2) or Microsoft Excel 2019 (version 2110)”
Results
Point 9: Line 241- by molecules do you mean antibiotic?
Response 9: Indeed, we mean antibiotic, correction have been made accordingly to ease reading as per suggested
Point 10: Table 2-7- please give the results for all the standard turbidity method and AST reference methods in the table so the reader can see where the differences were
Response 10: Thanks for this comment, we have made the change in all table according to the request
Point 11: Lines 345- 353 are repeated in lines 359-367
Response 11: Thank you for your careful revision, we have removed the duplication
Point 12: Table 3- I and S should not be italics
Response 12: Change was made accordingly
Point 13: Supplementary table S4-S7- please include full details similar to suggestion above for tables 2-7
Response 13: Thanks for this comment, we have made the change in all table according to the request
Discussion
Point 14: In the discussion you mention how good the test is at detecting ESBLs, CREs and MRSA, but there is no mention of these isolates in the methods or the results. Were they tested in this study and what were the results?
Response 14: We totally agree with your comment that detection of resistance mechanism are difficult to find within the different table, therefore we took the decision to create a specific table 8 and table S8 only describing the specific resistance and removing some of the data in the previous tables.
Point 15: What is the cost of the plates compared to disc diffusion?
Response 15: Thank you for this pertinent comment. However we are bound with a non-disclosure agreement for the moment that does not allow us to display this information widely to the public. We could provide the catalogue price of the microplate but to make the comparison possible and robust we would require adding in this article an entire chapter describing the cost between Microscan MSF panels and all related reagents and materiel versus standard methods; including MIC tests that are used to confirmed molecules that are not supposed to be tested by disc diffusion (Colistin, Vancomycin, Teicoplanin, etc.). We are preparing a manuscript where we will compare the cost between this method and conventional method for AST in LMIC taking in comparison the article by Roberts et all “ Antimicrobial resistance detection in Southeast Asian hospitals is critically important from both patient and societal perspectives, but what is its cost?” we therefore hope it will satisfy future readers
Point 16: Limitations section- another limitation is that you are comparing disc diffusion to MIC values. It is understandable that setting up E-tests or other MIC methods probably would not have been financially possible in these settings for comparison, but should just be mentioned in the limitations
Response 16: We thank you for this important comment and totally agree with it, we have added this indication in the limitation “Furthermore, comparing disc diffusion to MIC values is by itself a limitation of this study, but would have not been financially possible.”
Supplementary file
Point 17: Table S8- for isolate 135 results for BMD MIC-NEG1 and BMD Sensi-titre look wrong
Response 17: We have revised the results and make the change; accordingly, it has been a transcription error when creating the table
Reviewer 2 Report
The manuscript entitled “Validation of Three MicroScan® Antibiotic Susceptibility Testing (AST) Plates Designed for Low Resource Settings, written by Ronat et al is interesting. I enjoyed reading it. Following comments and suggestions should be considered before further consideration.
Abstract
The acronym “AST” used in the first sentence of the abstract should be written in full form.
Introduction
I would suggest that the authors add some information before going to the 1.1 subheading.
Add some information on the current status of AMR in LMICs with examples.
Line 4: LMIC should be replaced with “LMICs”
Line 67: The MSF should be written in full form.
Author Response
Dear Editor,
We would sincerely like to thank the reviewers for reading our manuscript and providing their valuable comments. In this letter, we will try to address some of their concerns. We hope we will be able to answer them in a satisfactory manner.
Reviewer # 2
Abstract
Point 1: The acronym “AST” used in the first sentence of the abstract should be written in full form.
Response 1: Thank you for the comment, we have made the change accordingly
Introduction
Point 2: I would suggest that the authors add some information before going to the 1.1 subheading. Add some information on the status of AMR in LMICs with examples.
Response 2: We do understand the importance of providing more information on AMR status in LMICs, however this research, also targeting improvement of AST for LMICs, focus on the description of lack of access to easy to use and standardized AST technics for those countries. Therefore, taking in consideration this information and the numerous numbers of articles rxisting and more recently the work done by Murray et all available to public, already describing in details AMR status in LMICs, we have only included in the introduction part a more generic approach with this sentence “Antimicrobial resistance (AMR) is today universally recognized as a global threat, because of the rapid emergence and dissemination of resistant bacteria and genes among humans, animals and the environment on a global scale and represents a heavy burden for healthcare systems all over the world especially in Low- and middle-income countries (LMICs)”
Point 3: Line 4: LMIC should be replaced with “LMICs”
Response 3: We have changed LMIC to LMICs as suggested
Point 4: Line 67: The MSF should be written in full form.
Response 4: Change has been made and MSF has been spelled in full form “Médecins Sans Frontières”